# Induction of Hepatoma Cell Pyroptosis by Endogenous Lipid Geranylgeranoic Acid—A Comparison with Palmitic Acid and Retinoic Acid

**DOI:** 10.3390/cells13100809

**Published:** 2024-05-09

**Authors:** Yoshihiro Shidoji

**Affiliations:** Graduate School of Human Health Science, University of Nagasaki, Nagayo, Nagasaki 851-2195, Japan; shidoji@sun.ac.jp

**Keywords:** acyclic retinoid, acyclic diterpenoid, cancer chemoprevention, lipotoxicity, palmitic acid, pyroptosis, regulated cell death, retinoic acid

## Abstract

Research on retinoid-based cancer prevention, spurred by the effects of vitamin A deficiency on gastric cancer and subsequent clinical studies on digestive tract cancer, unveils novel avenues for chemoprevention. Acyclic retinoids like 4,5-didehydrogeranylgeranoic acid (4,5-didehydroGGA) have emerged as potent agents against hepatocellular carcinoma (HCC), distinct from natural retinoids such as all-*trans* retinoic acid (ATRA). Mechanistic studies reveal GGA’s unique induction of pyroptosis, a rapid cell death pathway, in HCC cells. GGA triggers mitochondrial superoxide hyperproduction and ER stress responses through Toll-like receptor 4 (TLR4) signaling and modulates autophagy, ultimately activating pyroptotic cell death in HCC cells. Unlike ATRA-induced apoptosis, GGA and palmitic acid (PA) induce pyroptosis, underscoring their distinct mechanisms. While all three fatty acids evoke mitochondrial dysfunction and ER stress responses, GGA and PA inhibit autophagy, leading to incomplete autophagic responses and pyroptosis, whereas ATRA promotes autophagic flux. In vivo experiments demonstrate GGA’s potential as an anti-oncometabolite, inducing cell death selectively in tumor cells and thus suppressing liver cancer development. This review provides a comprehensive overview of the molecular mechanisms underlying GGA’s anti-HCC effects and underscores its promising role in cancer prevention, highlighting its importance in HCC prevention.

## 1. Introduction

Research on cancer prevention by retinoids has a long history. Reports of gastric cancer occurrence in animals deficient in vitamin A [1] and studies observing decreased serum vitamin A levels in patients with digestive tract cancer [2] were pioneering works in the early days following the discovery of vitamin A. Particularly notable is the study by Saffiotti [3], which demonstrated the suppression of chemical carcinogenesis in tracheal epithelium by excessive administration of vitamin A, laying the foundation for the concept of cancer chemoprevention by retinoids [4,5].

Significant research on the prevention of hepatocellular carcinoma (HCC) by retinoids includes the double-blind clinical trial conducted by Muto et al. using a placebo-controlled “acyclic retinoid” [6]. The administration of 4,5-didehydrogeranylgeranoic acid (4,5-didehydroGGA or peretinoin) at 600 mg/day for one year in patients after hepatic cancer resection significantly inhibited the occurrence of second primary liver cancer for up to five years post-administration. Subsequently, larger-scale clinical trials confirmed the hepatocellular carcinoma recurrence inhibitory effect of peretinoin [7,8].

Based on studies investigating the relationship between ligand activity on retinoid receptors and the differentiation-inducing effects on HCC cells, the concept of “acyclic retinoids” was proposed by Muto, Moriwaki, and Omori. Specifically, geranylgeranoic acid (GGA) and its derivative 4,5-didehydroGGA, which possess ligand activity for cellular retinoic acid-binding protein (CRABP) [9] and retinoid receptors (RAR and RXR) [10], as well as differentiation-inducing activity on HCC cell lines [10], were defined as acyclic retinoids. On the other hand, we demonstrated that GGA, unlike all-*trans* retinoic acid (ATRA), is an endogenous fatty acid synthesized from mevalonic acid in mammalian hepatocytes [11]. Therefore, although GGA is an acyclic retinoid, it cannot be considered a vitamin.

GGA and ATRA belong to the long-chain polyunsaturated branched-chain fatty acids with four to five unsaturated bonds and a carbon number of 20. On the other hand, palmitic acid (PA), a long-chain saturated fatty acid, is well known for inducing cell death as a lipotoxic effect on various cells, along with cholesterol and lysophospholipids [12]. Interestingly, PA is also a fatty acid considered for HCC prevention. However, when the hypoxia-dependent cell death-inducing effect of PA on HCC cells was first discovered, it was believed to be related to the pathogenesis of non-alcoholic fatty liver disease [13]. However, recently, studies proposing the application of PA-induced cell death for cancer treatment have also been reported [14].

Therefore, in this review, we first summarize the inhibitory effect of GGA on HCC as an induction of cell death in HCC cells from the perspective of pyroptosis. Subsequently, we compare the differences in the cell death-inducing effects of GGA, ATRA, and PA on HCC cells based on the latest literature and discuss the molecular targets of GGA on HCC cells.

## 2. Mechanism of Cell Death Induction by GGA in Hepatocellular Carcinoma (HCC) Cells

GGA, similar to ATRA, has been confirmed to induce a significant increase in the expression of neurotrophic receptor kinase 2 (NTRK2 or TrkB) and RARβ protein in human neuroblastoma cell line SH-SY5Y, indicating that GGA possesses a “retinoid effect” like ATRA [15]. However, the cell death-inducing effect of GGA on HCC cells, as described below, is a unique action of GGA that is not found in natural retinoids such as ATRA.

Given the detailed historical background of research on the cell death induction mechanism of GGA in HCC cells [11], we will summarize all the findings obtained to date and present the proposed mechanism.

### 2.1. Timeline of Intracellular Events Related to Cell Death in Human Hepatocellular Carcinoma (HCC)-Derived Cell Lines Induced by GGA (Figure 1)

Describing the intracellular events occurring in HCC cells after GGA treatment along a timeline can provide an approximate mechanism of GGA-induced cell death. Of course, the observation of events depends on the analysis technique (detection sensitivity, complexity of the technique, etc.), so the actual sequence may sometimes be reversed. Below, we describe the phenomena observed over time after the treatment of human HCC-derived cell lines (mainly HuH-7) with GGA (10–20 µM):15 min: The earliest observed phenomena upon adding 10 µM GGA to the culture medium include three main events. Firstly, increased superoxide production in mitochondria is observed using the MitoSox fluorescence staining technique [16]. Secondly, the splicing of *XBP1* mRNA in the endoplasmic reticulum (ER) is detected using the RT-qPCR technique [17,18]. Thirdly, an increase in LC3β-II protein, a marker of autophagosomes, is observed using Western blotting [16]. Although these events are observed simultaneously in different cellular organelles (mitochondria, ER, and autophagosomes), it is unlikely that these events occur independently and simultaneously. Since suppressing the increased production of superoxide by GGA with antioxidants does not suppress *XBP1* mRNA splicing [18], it appears that at least the increased production of superoxide does not trigger *XBP1* mRNA splicing.20 min: Monitoring the cytosolic Ca^2+^ concentration using Fluo-4 AM as a probe reveals a transient peak 16–20 min after GGA treatment, which quickly declines ([18]; see Appendix A). This peak is observed, albeit slightly delayed, even when experiments are conducted in a medium without Ca^2+^, suggesting the leakage of Ca^2+^ from intracellular Ca^2+^ storage sites such as the ER.30 min: Introduction of the GFP-LC3 expression vector into HuH-7 cells followed by GGA treatment results in the appearance of green, fluorescent autophagosomes 30 min later [16], consistent with the detection of the LC3β-II protein 15 min earlier. An increase in Beclin-1 (BECN1) and Sequestosome 1 (p62/SQSTM1), which are involved in promoting autophagosome formation and a cargo protein for autophagosomes, respectively, is also detected [16]. Meanwhile, although there is no change in the transcription level of the *Cyclin D1* (*CCND1*) gene, its translation is significantly suppressed [19].1 h: Loss of the mitochondrial membrane potential (ΔΨ*m*) is observed 1 h after GGA treatment when stained with Rhodamine 123 [20] and 2 h after GGA treatment when stained with MitoTracker^®^ Red CMXRos (Thermo Fisher Scientific, Tokyo, Japan) [16]. Regardless of the staining method, the loss of ΔΨ*m* is observed after the increased superoxide production in mitochondria. Active caspase-4 (CASP4) is detected by Western blotting and observed up to 5 h after the GGA addition but disappears thereafter [18]. Concurrently with the detection of active CASP4, the N-terminal fragment of Gasdermin D (GSDMD) is detected, with its amount peaking at 3 h and then decreasing, reaching its maximum at 8 h after GGA treatment when cell death is observed.2 h: A transient increase in lysophosphatidylcholine containing PA and palmitoleic acid (lysoPC [16:0]; lysoPC [16:1]) is observed. In contrast, a significant increase in lysoPC (lysoPC [20:4]) and lysophosphatidylethanolamine (lysoPE [20:4]) containing arachidonic acid is observed, and then, these lysophospholipids (lysoPLs) remain at high levels and continue to increase gradually until 24 h [21].3 h: Immunofluorescence staining of GSDMD reveals signals mainly in the nucleus in control cells, but in GGA-treated cells, the signals are also detected in the plasma membrane [18]. Immunofluorescence staining of NF-κB, one of the inflammatory transcription factors, shows that its signal is observed granularly in the cytoplasm in control cells but most of it translocates into the nucleus after GGA treatment [18]. Morphological changes in HuH-7 cells are first observed, with loss of cell adhesion and contraction of cells away from each other. Rod-shaped protrusions or blebs emerge from each contracted cell into the intercellular space ([18]; see Appendix A).6 h: The transient peak in the cytosolic Ca^2+^ concentration observed 20 min after GGA treatment reappears 6 h later. When experiments are conducted in a medium without Ca^2+^, this peak is completely absent, suggesting that it is due to a Ca^2+^ influx into the cytosol from the medium through a perforated cell membrane by GSDMD translocated to the membrane. The intracellular levels of *NLRP3* mRNA and *IL1B* mRNA increase. Blebs protruding from contracted cells disappear, and spherical balloons appear, gradually increasing in size [18].8 h: Significant activation of CASP1 is observed by enzyme activity measurement. Detection of the N-terminal fragment of GSDMD also peaks. Leakage of lactate dehydrogenase (LDH) into the medium is observed. The balloons that emerge from contracted cells become larger than the cell diameter [18]. (For chronological morphological changes, [18]; see Appendix A).
Figure 1Timeline of cellular events detected after GGA treatment in HuH-7 cells. 
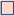
: mitochondria-related, 
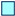
: UPR^ER^-related, 
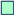
: autophagy-related, 
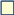
: lysoPLs-related, and 
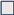
: pyroptosis-related. One down-blue arrow indicates downregulation, and all other boxes indicate upregulation. Details of this diagram are given in the text.
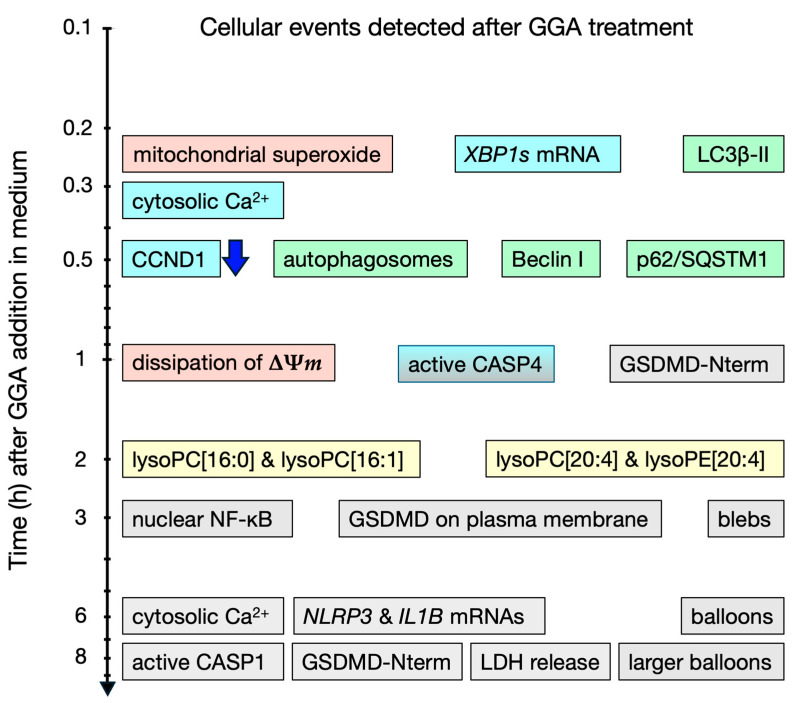



### 2.2. Inhibition of GGA-Induced Cell Death by Various Inhibitors

CASP Inhibitory Peptides: The induction of cell death in HuH-7 cells by GGA is completely inhibited by cotreatment with an active site inhibitor peptide of CASP1 (ac-YVAD-cmk). Cotreatment with an active site inhibitor peptide of CASP3 (ac-DEVD-CHO) delays GGA-induced cell death by several hours but does not inhibit it [18]. Cotreatment with an active site inhibitor peptide of CASP4 (Z-LEVD-fmk) inhibits the GGA-induced activation of CASP1 [18].TLR4 Inhibitors: The pretreatment of HuH-7 cells with *TLR4* (Toll-like receptor 4) siRNA completely inhibits GGA-induced cell death. Cotreatment with VIPER, a peptide that specifically inhibits signaling from TLR4, completely inhibits GGA-induced cell death, as well as other processes induced by GGA, such as increased superoxide production in mitochondria, the splicing of *XBP1* mRNA, nuclear translocation of the cytosolic NF-κB, upregulation of *TLR2* and *NLRP3* mRNAs, and activation of CASP1 [18].Lipotoxicity Inhibitor: Cotreatment with oleic acid (OA), known to inhibit PA-induced lipotoxicity, completely inhibits GGA-induced cell death in HuH-7 cells. Additionally, OA cotreatment inhibits various processes induced by GGA, including the splicing of *XBP1* mRNA, upregulation of *TLR2* mRNA and *DDIT3* (*CHOP*) mRNA, increase in LC3β-II levels, accumulation of autophagosomes, superoxide hyperproduction in mitochondria, translocation of cytoplasmic NF-κB to the nucleus, and activation of CASP1 [18]. The inhibitory effect of OA on GGA is observed only when co-treated simultaneously; treatment with OA before GGA treatment does not exhibit any inhibitory effect [17], suggesting that the inhibitory point of action of OA on GGA is extracellular.Antioxidant: Pretreatment with α-tocopherol, a lipid-soluble antioxidant vitamin, dose-dependently inhibits GGA-induced cell death [20]. Cotreatment with α-tocopherol also inhibits other processes induced by GGA, such as the dissipation of ΔΨ*m*, activation of CASP1, upregulation of *NLRP3* mRNA, translocation of cytoplasmic NF-κB to the nucleus, and increased superoxide production in mitochondria [17,18]. However, cotreatment with α-tocopherol does not inhibit activation of an unfolded protein response in the ER (UPR^ER^: splicing of *XBP1* mRNA and upregulation of *DDIT3* mRNA) by GGA [18].Kinase Inhibitors: Cotreatment with wortmannin, a non-specific inhibitor of PI3 kinase, inhibits a GGA-induced increase in superoxide production in mitochondria and the appearance of green, fluorescent puncta in GGA-treated GFP-LC3-transfected HuH-7 cells [16]. Each cotreatment with BAY 11 7082, an inhibitor of NF-κB activation, or BI605906, an IKKβ inhibitor, inhibits the translocation of cytoplasmic NF-κB to the nucleus induced by GGA [18].

### 2.3. The Mechanism of GGA-Induced Cell Death

Based on the timeline of intracellular events after GGA treatment described above, along with observations of the effects of various inhibitors, the mechanism of cell death induction by GGA in HCC cells is currently understood as follows (Figure 1).

The inhibition of *TLR4* gene expression using specific siRNA or inhibition of intracellular signaling of TLR4 by the virus-derived peptide VIPER suppresses GGA-induced cell death. Cotreatment with VIPER affects almost all processes associated with GGA-induced cell death. These include the enhanced production of superoxide in mitochondria, activation of UPR^ER^, translocation of cytoplasmic NF-κB to the nucleus, priming of the inflammasome (upregulation of *NLRP3* and *IL1B* mRNAs), localization of GSDMD to the cell membrane, and activation of CASP1 [18], suggesting that TLR4 serves as the sole initiator of GGA-induced cell death.

The signal of TLR4 stimulated by GGA appears to bifurcate into two major pathways within the cell. One involves the UPR^ER^, detected as the splicing of *XBP1* mRNA and upregulation of *DDIT3* mRNA, along with the consequent activation of ER resident CASP4, leading to the formation of GSDMD pores through the cleavage of GSDMD by active CASP4 and the translocation of the GSDMD N-terminal fragment to the cell membrane (Figure 2, left). Although the precise mechanism linking TLR4 signaling to UPR^ER^ is not yet clear, several reports suggest that the stimulation of TLR4 induces UPR^ER^, particularly the splicing of *XBP1* mRNA [22,23,24,25].

It is believed that the mechanism by which UPR^ER^ activates CASP4 involves the release of Ca^2+^ from the ER. Indeed, treatment solely with thapsigargin, which decreases the Ca^2+^ concentration in the ER by inhibiting the sarcoplasmic/endoplasmic reticulum Ca^2+^ ATPase known as an ER stress inducer, activates CASP4 similarly to GGA treatment [18]. Hitomi et al. [26] first reported the activation of CASP4 by UPR^ER^, and since then, the activation of ER resident CASP4 via UPR^ER^ has been confirmed in numerous studies [27,28,29,30]. Recently, it has been demonstrated that calpain 5 (CAPN5), localized in mitochondria, is activated by ER stress, and cleaves pro-CASP4 into its active form, serving as a mechanism for CASP4 activation [31]. Conversely, treatment with tunicamycin, an inhibitor of protein N-glycosylation and another ER stress inducer, alone did not activate CASP4 [18]. Therefore, GGA may activate CASP4 through a putative calpain (maybe CAPN5) activated by the initial Ca^2+^ peak at 16–20 min [18]. Activated CASP4 cleaves GSDMD to generate N-terminal fragments, which translocate to the cell membrane and form GSDMD pores. The translation inhibition of the *CCND1* gene observed 30 min after GGA treatment [19] is also considered a downstream signal of UPR^ER^ [32].

The other pathway of TLR4 signaling involves the enhancement of superoxide production in mitochondria (Figure 2, right). Upon TLR4 activation, the TLR4 adapter protein TRAF6 (TNF receptor-associated factor 6) translocates to the mitochondria and interacts with ECSIT (Evolutionarily conserved signaling intermediate in Toll pathway, mitochondrial), promoting its ubiquitination. ECSIT, a subunit of the Mitochondrial respiratory chain Complex I Assembly complex (MCIA), is thought to be one of the chaperones for complex I [33,34,35]. When ECSIT is ubiquitinated, MCIA undergoes degradation, leading to the increased production of reactive oxygen species (ROS) in mitochondria [36]. Excessive ROS generated in mitochondria promote the translocation of cytoplasmic NF-κB to the nucleus, increasing the transcription of *NLRP3* and *IL1B* genes (priming of inflammasomes). Activation of the NLRP3 inflammasome by extracellular Ca^2+^ influx through GSDMD pores formed by CASP4 leads to the activation of CASP1 [37], which, in turn, cleaves GSDMD, resulting in the formation of GSDMD pores in the plasma membrane and subsequent pyroptotic cell death.

In summary, TLR4 signaling plays an essential role in GGA-induced cell death, and UPR^ER^-associated activation of CASP4 occurs within 1 h after GGA addition. The cleavage of GSDMD leading to the formation of GSDMD-N-terminal fragments and their localization to the plasma membrane is observed at 3 h, suggesting activation of the non-canonical pathway of pyroptosis [18]. However, cell death is not observed at 3 h after GGA addition. Subsequently, activation of CASP1 by the NLRP3 inflammasome is observed, along with the dramatic upregulation of *IL1B* gene expression, suggesting the possibility of the activation of canonical pathway-mediated pyroptosis [18]. Regarding this, the inhibition of CASP1 activation by cotreatment with a CASP4 inhibitor during GGA treatment suggests that activation of the non-canonical pathway may trigger the activation of the canonical pathway. Indeed, recent reviews [38,39] suggesting that the non-canonical pathway’s efflux of K^+^ to the extracellular space and increased mitochondrial ROS production serve as triggers for the secondary activation of the canonical pathway support our findings.

However, attributing GGA’s cell death induction solely to TLR4 signaling is insufficient. For instance, the upregulation of lysophospholipids or incomplete autophagy response (possibly due to lysosome inactivation) induced by GGA might occur via alternative pathways independent of authentic TLR4 signaling, playing essential roles in GGA-induced cell death. Further detailed research is required for elucidating the molecular role of GGA.

## 3. GGA-Induced Cell Death in Hepatocellular Carcinoma (HCC) Cells—A Comparison with Palmitic Acid (PA)-Induced Cell Death and All-*trans* Retinoic Acid (ATRA)-Induced Cell Death

GGA is a branched-chain fatty acid derived from mevalonic acid, with 20 carbon atoms and four unsaturated bonds [11]. PA, a saturated fatty acid with 16 carbon atoms, is well known for its lipotoxicity to cells such as hepatocytes [12]. While the molecular lengths of both molecules are almost the same, the major difference lies in the presence of unsaturated bonds and branched chains. On the other hand, ATRA, shown in the lower panel of Figure 3, is a 20-carbon, branched-chain fatty acid with five unsaturated bonds and is believed to be one of the active forms of vitamin A, converted in the body from β-carotene and vitamin A esters found in food. ATRA has also been reported to induce cell death in HCC-derived cell lines [40,41]. Therefore, comparing the differences between cell death induced by GGA in HCC-derived cell lines and that induced by PA or ATRA, we can evaluate the significance of GGA-induced cell death.

### 3.1. Effective Concentrations and Observation Time for Inducing Cell Death (Table 1)

Since the targeted cell lines and experimental conditions vary, a direct comparison may not be meaningful, but citations using human HCC cell lines are compared as much as possible. Experiments inducing cell death in human HCC cell lines by GGA were conducted at final concentrations of 0.5–20 µM, and cell death was observed within 8 to 16 h after GGA addition.

For ATRA, the concentrations are added in the medium range from 0.1 to 20 µM, like GGA, but it does not always induce cell death. Although we conducted experiments under the same conditions as with GGA, no cell death was observed in ATRA-treated cells even after 24 h [42]. Lin et al. reported that treatment with 0.1–10 µM ATRA for 3 days did not affect the proliferation of HepG-2 and Hep3B cells [43], and Wang et al. found that 5–20 µM ATRA for HepG-2 and HuH-7 cells and 1–5 µM ATRA for Hep3B cells protected them against cell death occurring 5–10 days after serum removal [44]. Although reports of ATRA treatment inhibiting cell proliferation exist, significant effects are observed after 24 h to more than 5 days [40,41].

On the other hand, there are numerous reports of cell death induction by PA, mainly in the analysis of lipotoxicity to immune cells. Here, only reports using human HCC-derived cell lines are discussed. The final concentrations of PA are added in the medium range from 200 to 3000 µM, which is ten to several hundred times higher than GGA or ATRA. Although there are many reports, cell death is confirmed within 24 h after PA treatment [13,14,45,46,47,48]. Molecules that exert effects at concentrations of several µM like GGA or ATRA are thought to act as signaling molecules. In contrast, PA, which only exerts effects at concentrations of several hundred µM to several mM, may have its metabolites acting as effector molecules rather than itself being a signaling molecule. In other words, it may be possible to consider the cell death induction effect of PA because of changes in the membrane fluidity or compactness caused by its metabolites.

In summary, cell death induced by GGA occurs at considerably lower concentrations and more rapidly compared to PA and even more rapidly compared to ATRA.

### 3.2. Mode of Regulated Cell Death

All three fatty acids—GGA, PA, and ATRA—appear to induce cell death in HCC-derived cells accompanied by the activation of CASP3, as shown in Table 1. However, the typical intrinsic or extrinsic mechanisms of apoptosis have not been consistently observed. Activation of NF-κB, CASP1, and CASP4 has been observed with GGA treatment, and activation of NF-κB and CASP1 has been noted with PA treatment, thus suggesting the involvement of pyroptosis mechanisms in cell death induction. Conversely, ATRA treatment has been reported to increase the expression of the *OTUD7B* gene, an NF-κB inhibitory factor [49], indicating that inflammatory cell death, as observed with GGA or PA treatment, is unlikely to occur. ATRA has been shown to induce apoptotic cell death in human HCC cell lines PLC/PRF/5 and HLE [50] and ferroptosis cell death in HepG2 and Hep3B cells [51]. However, there are also reports suggesting that ATRA inhibits ferroptosis in neuronal cell lines [52].

### 3.3. Action on Plasma Membrane and Intracellular Organelles

Plasma Membranes: When fatty acids like GGA, PA, and ATRA are added to the medium, they are expected to encounter the plasma membrane. Both PA and GGA are believed to induce cell death via signaling mediated by the cell surface receptor TLR4, which is localized on the plasma membrane [18,53]. However, few reports have suggested that these lipids directly act as ligands for TLR4. While computational simulations and isothermal titration calorimetry have demonstrated the docking of five PA molecules with the hydrophobic pocket of TLR4/MD-2, experimental evidence confirming PA as a direct ligand for TLR4 is lacking [54].

On the other hand, certain lipids are known to activate TLR4 through lipid rafts on the cell membrane. Lipid rafts are enriched with saturated lipids (such as sphingomyelin and glycerophospholipids containing saturated fatty acids) and cholesterol. The addition of these lipids increases the lipid phase order, leading to the dimerization of TLR4 and enhanced signaling [55]. Therefore, it is speculated that the increase in glycerophospholipids containing saturated fatty acids, such as PA, upon PA addition may lead to the activation of TLR4 through lipid rafts. However, unlike lipopolysaccharide (LPS), a known ligand for TLR4/MD-2, treatment with PA does not induce TLR4 dimerization or internalization, as observed with LPS treatment [53].

Since GGA is not known to be incorporated into sphingomyelin or glycerophospholipids, it might be inserted into the plasma membrane as unchanged GGA. If we assume that GGA micelles prepared in the medium fuse with the lipid bilayer of the plasma membrane and GGA is inserted into lipid rafts through flip-flops in the bilayer, GGA might promote an increase in phase order of the plasma membrane, like cholesterol, as an isoprenoid, leading to the activation of TLR4 through lipid rafts. However, further experimental studies are essential to establish this. There are currently no reports suggesting that TLR4 is essential for ATRA-induced cell death in HCC cells or that ATRA activates hepatic TLR4.

Mitochondria: The effects on mitochondrial morphology and function are relatively similar among GGA, PA, and ATRA. Adding GGA to cultured HCC-derived cell lines leads to mitochondrial accumulation around the nucleus and fragmentation within 2 h [16]. Similarly, mitochondrial fragmentation is observed within 2 h of PA treatment [56]. ATRA treatment results in a decreased mitochondrial number and average volume, as well as the disappearance of mitochondrial cristae after 48 h [51]. Additionally, the loss of ΔΨ*m* is observed after 1 h of GGA (10 µM) treatment [20], 24–96 h of ATRA (1 µM) treatment [57], and 24 h of PA (200 µM) treatment [48], indicating mitochondrial permeability transition (mPT) (Table 1). Furthermore, the hyperproduction of superoxide in mitochondria is confirmed after treatment with GGA (10µM, 15 min) [16], PA (500 µM, 3–24 h) [48,58], or ATRA (10 µM, 48 h) [51]. Free fatty acids like PA are proposed to induce superoxide production by directly binding to complexes I and III of the respiratory chain, inhibiting electron transfer and leading to increased superoxide production [59]. Although the signaling from TLR4 to ECSIT resulting in an impaired electron transport chain in mitochondria has been proposed for GGA (as mentioned earlier), the mechanism of increased superoxide production by ATRA remains unclear in the literature.Endoplasmic reticulum (ER): UPR^ER^ is a common response of HCC cells to treatment with these three fatty acids. Treatment with GGA (10–20 µM, 15–30 min) [17,18], PA (400–500 µM, 8–24 h) [46,58,60], or ATRA (10–20 µM, 1–8 h) [17] results in the splicing of *XBP1* mRNA, translocation of XBP1 protein to the nucleus, and increased expression of *DDIT3* (*CHOP*) mRNA. The induction of UPR^ER^ by ATRA has also been observed in mouse embryonal carcinoma cell line P19, not limited to HCC cells [61]. Membrane phospholipids derived from PA are believed to affect the fluidity of the ER membrane and induce UPR^ER^ through the dimerization of IRE1 [62]. Since effective concentrations of GGA and ATRA are in the order of 10 µM and induce relatively rapid responses, they are likely mediated through signaling, although detailed mechanisms require further investigation.Nucleus: The translocation of NF-κB to the nucleus due to the increased production of ROS is a well-known phenomenon [63]. Indeed, treatment with GGA induces the translocation of NF-κB to the nucleus, which can be inhibited by cotreatment with the antioxidant α-tocopherol or suppression of increased superoxide production in the mitochondria [18]. Additionally, GGA treatment promotes the rapid translocation of the cytoplasmic p53 to the nucleus and increases expression of the *PUMA* gene, one of the p53 target genes [64]. Like GGA, PA also induces nuclear NF-κB activity in HepG2 [65,66] and HepaRG cells [66].Cytosol: An increase in the cytosolic Ca^2+^ concentration is another common response of HCC cells to treatment with these three fatty acids (Table 1). However, the effective concentrations and time of occurrence of this effect differ. Treatment with 10 µM of GGA leads to two increases in the cytosolic Ca^2+^ concentration at 20 min and 6 h [18], while PA requires concentrations of 500–1000 µM, with an increase observed 6 h after treatment [48]. Similarly, ATRA requires 24–48 h after treatment to observe an increase in the cytosolic Ca^2+^ concentration, despite effective concentrations being in the order of 10 µM [67]. The mechanism of an increased cytosolic Ca^2+^ concentration by PA has been most extensively analyzed. PA enhances mitochondrial ROS (mtROS) production, which releases Ca^2+^ from lysosomes, resulting in an increased cytosolic Ca^2+^ concentration and leading to mPT and cell death [48].Inflammatory extracellular vesicles: When hepatocytes and HCC cells are treated with PA, they secrete inflammatory extracellular vesicles [68,69]. It is believed that lysoPC, one of the intracellular metabolites of PA, is directly involved in the release of inflammatory extracellular vesicles [70]. It has been reported that one-tenth to one-twentieth lysoPC induces cell death within 24 h in primary human hepatocytes to the same extent as cell death by PA [71]. Considering that lysoPC increases upon GGA treatment of HCC cells, it is hypothesized that GGA treatment may lead to the release of inflammatory extracellular vesicles. In contrast, since lysoPC is reduced upon ATRA treatment, the release of extracellular vesicles is unlikely after ATRA treatment [72].

### 3.4. Role of Mitochondrial ROS Hyperproduction, UPR^ER^, and Autophagy in GGA-Induced Cell Death

As mentioned earlier, the effects of the three fatty acids on human HCC cells can be broadly categorized into two types: pyroptosis induced by GGA and PA and apoptosis induced by ATRA. Regarding increased mitochondrial ROS (mtROS) production and UPR^ER^, similar phenomena occur with all three fatty acids. However, the most significant difference lies in the occurrence of autophagy. In the case of ATRA treatment, there is an increase in the flux of autophagy, accompanied by an increase in LC3-II and degradation of p62/SQSTM, which is the cargo of autophagosomes [50]. Moreover, intriguingly, the knockdown of the essential autophagy gene ATG7 leads to the induction of apoptosis by ATRA [50].

On the other hand, while the marker of autophagosomes, LC3-II increases with GGA and PA treatment, autophagic flux is inhibited, and p62/SQSTM increases, opposite to ATRA [16,73]. Data indicating the abnormal accumulation of early autophagosomes and impaired formation of autolysosomes due to GGA treatment suggest an “incomplete response” of autophagy (or impaired autophagy) is involved in GGA-induced cell death [16]. Subsequently, autophagy has been shown to suppress pyroptosis. In other words, autophagy inhibits the formation of canonical and non-canonical inflammasomes by sequestering damaged mitochondria generating ROS, and Gram-negative bacteria into autophagosomes, which are then cleared via autolysosomes [74]. Therefore, in the cases of GGA and PA treatment, incomplete autophagic responses or the cessation of autophagy progression may lead to the inability to regulate inflammasomes, resulting in the execution of cell death by pyroptosis. Recent studies have reviewed the involvement of incomplete autophagic responses in PA lipotoxicity in hepatocytes [75] and adipocytes [76] other than HCC cells. In neurons, incomplete autophagic responses induced by PA are reported to lead to decreased insulin sensitivity [77].

How is autophagy progression impaired by GGA or PA treatment? Reports suggest that the treatment of macrophages primed by LPS, a TLR4 ligand, with PA induces lysosomal dysfunction, which plays a crucial role in the lipotoxicity of PA [78]. Reports of decreased Ca^2+^ concentrations in lysosomes upon PA treatment [48] are intriguing in this regard. MtROS activate the Ca^2+^ efflux channel of lysosomes. Since the cytosolic Ca^2+^ concentration increases with GGA treatment and autophagy halts at the stage of autophagosomes, the increase in cytosolic Ca^2+^ concentration is presumed to be related to lysosomal dysfunction. However, since mtROS production is also increased by ATRA treatment, it is unlikely that lysosomal dysfunction is caused solely by mtROS.

Additionally, Karasawa et al. [79] reported that lysosomal dysfunction caused by PA is due to the precipitation of PA-derived crystals within cells and that lysosomal dysfunction induces inflammasome activation. Considering that the melting point (62.9 °C) of PA is much higher than body temperature, they may be crystals of the PA itself. Cotreatment of PA with oleic acid, which has a much lower melting point (13.4 °C), prevents the observation of intracellular crystals. However, the phenomenon of intracellular crystallization by PA treatment is observed in primary cultures of macrophages or macrophage-like cell lines but not in hepatocellular carcinoma cell lines [79]. Moreover, since GGA remains liquid even at room temperature, it is unlikely that GGA crystallizes within living cells after GGA treatment.

In any case, in GGA or PA treatment, the inhibition of autolysosome formation due to lysosomal dysfunction results in an incomplete autophagic response, causing autophagy to halt midway, and consequently, control over pyroptosis is lost.

**Table 1 cells-13-00809-t001:** Intracellular events of GGA-induced cell death in HCC cells vs. PA and ATRA.

	GGA	PA	ATRA
Cytotoxicity	➢ dGGA (0.5–10 µM) induced cell death (trypan blue) in HuH-7 cells in 16 h (Nakamura 1995) [42]. ➢ GGA (10 µM) induced cell death (trypan blue) in HuH-7 cells at 16 h, but not in normal hepatocytes (Shidoji 1997) [20]. ➢ Cell death in HuH-7 cells induced by 10 µM GGA was prevented by cotreatment with 100 µM α-tocopherol (Shidoji 1997) [20]. ➢ Cell death in HuH-7 cells induced by 50 µM GGA was prevented by cotreatment with 25 µM oleic acid (OA) (Iwao 2015) [17]. ➢ Cell death in HuH-7 cells induced by 20 µM GGA was prevented by pretreatment with *TLR4* siRNA or VIPER, a TLR4 inhibitor (Yabuta 2020) [18].	➢ PA (500–3000 µM) induced cell death (MTT) in HepG2 cells in 24 h (Saha 2022) [47]. ➢ PA (500–1000 µM) induced cell death (SYTOX Green) in HepG2 cells in 24 h, which was suppressed by cotreatment with the mitochondrial ROS quencher, MitoTEMPO, or the mPT inhibitor, olesoxime (Oh 2023) [48]. ➢ PA (200 & 400 µM) induced cell death in HepG2 cells in 24 h (Li 2021) [45], which was suppressed by cotreatment with OA (Zeng 2020) [46]. ➢ PA (1000 and 2000 µM) induced hypoxia-dependent cell death in HepG2 cells in 24 h (Hwang 2015) [13]. ➢ PA (200 µM) induced hypoxia-dependent cell death in HepG2 and Hep3B cells in 48 h, which was enhanced by cotreatment with L-carnitine (Matsufuji 2023) [14]. ➢ PA (1000 µM) induced TLR4-dependent inflammation in primary macrophages in 4 h (Lancaster 2018) [53].	➢ ATRA (10–20 µM) induced no cell death in HuH-7 cells in 24 h (Nakamura 1995) [42]. ➢ ATRA (1–10 µM) decreased cell viability (trypan blue) in Hep3B cells (Hsu 1998) [40]. ➢ ATRA (10 µM) inhibited the proliferation of Hepa1-6 and HepG2 cells in 5 days (Fang 2020) [41]. ➢ ATRA (5–20 µM) protected HepG2 or HuH-7 cells and 1–5 µM ATRA protected Hep3B cells from serum starvation-induced cell death in 5–10 days (Wang 2013) [44]. ➢ ATRA (0.1–10 µM) did not inhibit the proliferation of HepG2 or Hep3B cells in 3 days (Lin 2005) [43].
Mitochondrial morphology	➢ GGA (10 µM) induced in HuH-7 cells in 2 h, − a perinuclear clustering of mitochondria (Okamoto 2011) [16]. − a fragmentation of mitochondria (unpublished, Shidoji).	➢ PA (200 µM) induced mitochondrial fragmentation in HepG2 cells in 2 h (Eynaudi 2021) [56].	➢ ATRA (10 µM) in HepG2 and Hep3B cells in 48 h, − reduced the number of mitochondria (Sun 2022) [51]. − reduced mitochondrial volume (Sun 2022) [51]. − reduced or eliminated mitochondrial cristae (Sun 2022) [51]. ➢ ATRA (1 µM) did not alter the mitochondrial mass for 24–72 h in the acute-promyelocytic-leukemia (APL)-derived NB4 cell line (Gianni 2022) [57].
Mitochondrial membrane potential (ΔΨm)	➢ GGA (10 µM) induced in HuH-7 cells, − a dissipation of Rhodamine 123 fluorescence 1 h, which was prevented by cotreatment with 100 µM α-tocopherol (Shidoji 1997) [20]. − a dissipation of MitoTracker Red fluorescence in 2 h (Okamoto 2011) [16].	➢ PA (500 µM) induced dissipation of TMRE red fluorescence (mitochondrial permeability transition or mPT) in HepG2 cells in 24 h, which was abrogated by cotreatment with BAPTA-AM chelating intracellular Ca^2+^ (Oh 2023) [48].	✧ ATRA (1 µM) reduced Mito-Tracker Red/Green fluorescence in NB4 cells for 24–96 h (Gianni 2022) [57].
Mitochondrial ROS production	➢ GGA (10 µM) upregulated in HuH-7 cells, − MitoSox Red fluorescence of hepatoma cells in 15 min, which was prevented by cotreatment with wortmannin (Okamoto 2011) [16]. − MitoSox Red fluorescence of hepatoma cells in 2 h, which was prevented with either 50 µM oleic acid (OA), 100 µM α-tocopherol or VIPER (Yabuta 2020) [18].	➢ PA (500–1000 µM) upregulated MitoSox fluorescence in HepG2 cells in 24 h, which was suppressed by cotreatment with MitoTEMPO, a mitochondrial ROS quencher (Oh 2023) [48]. ➢ PA (500 µM) increased ROS production at 3 h in both mouse primary hepatocytes and mouse hepatocyte-derived AML12 cells (Huang 2023) [58].	➢ ATRA (10 µM) upregulated in HepG2 and Hep3B cells in 48 h, − H2-DCF-DA fluorescence (Sun 2022) [51]. − lipid peroxides (C11-BODIPY fluorescence) (Sun 2022) [51].
ER stress response(UPR^ER^)	➢ GGA (10–20 µM) induced in HuH-7 cells, − translational downregulation of cyclin D1 (CCND1) in 30 min (Shimonishi 2012) [19]. − splicing of *XBP1* mRNA in 15 min (Iwao 2015) [17], which was prevented by cotreatment with either 50 µM OA or VIPER, but not with 100 µM α-tocopherol (Yabuta 2020) [18]. − accumulation of XBP1 protein in the nucleus in 8 h (Iwao 2015) [17]. − upregulation of *DDIT3* (*CHOP*) mRNA in 8 h (Iwao 2015) [17], which was prevented by cotreatment with 50 µM OA or VIPER, but not with 100 µM α-tocopherol (Yabuta 2020) [18].	➢ PA (500 µM) induced in HuH-7 cells, − splicing of *XBP1* mRNA in 16 h (Qi 2015) [60]. − upregulation of *DDIT3* (*CHOP*) protein in 16 h (Qi 2015) [60]. − phosphorylation of IRE1α in 8 h (Qi 2015) [60]. ➢ PA (400 µM) enhanced expressions of *DDIT3* (*CHOP*) mRNA after 24 h in HepG2 cells, which was prevented by cotreatment with 200 µM OA (Zeng 2020) [46]. ➢ PA (500 µM) enhanced cellular expression of p-PERK, ATF4, p-eIF2α, DDIT3 (CHOP), and TXNIP in both mouse primary hepatocytes and mouse hepatocyte-derived AML12 cells (Huang 2023) [58].	➢ ATRA (10–20 µM) induced splicing of *XBP1* mRNA in 1 h in HuH-7 cells (Iwao 2015) [17]. ➢ ATRA (10–20 µM) induced accumulation of XBP1 protein in the nucleus of HuH-7 cells in 8 h (Iwao 2015) [17]. ✧ In P19 embryonic carcinoma cells, ATRA induced the upregulation of several UPR-related genes (*Atf6*, *Xbp1*, *Chop*) (Saito 2023) [61].
Pyroptosis	➢ GGA (20 µM) induced in HuH-7 cells, − nuclear translocation of cytoplasmic NF-κB in 3 h, which was blocked by cotreatment with 50 µM OA, 100 µM α-tocopherol, BI605906 or VIPER (Yabuta 2020) [18]. − upregulation of TLR2 expression in 3 h, which was prevented with 50 µM OA or VIPER (Yabuta 2020) [18]. − upregulation of *NLRP3* expression in 3 h, which was prevented with 50 µM OA, 100 µM α-tocopherol or VIPER (Yabuta 2020) [18]. − appearance of GSDMD-N terminal fragment in 1 h (Yabuta 2020) [18]. − localization of GSDMD to the plasma membrane in 3 h, which was prevented by cotreatment with VIPER (Yabuta 2020) [18]. ➢ Cell death induced by 10 µM GGA was completely prevented by pre-treatment with acYVAD-cmk, a specific inhibitor against CASP1 (Shidoji 1997) [20]. ➢ GGA (20 µM) induced activation of CASP1 activity in 8 h, which was prevented by Z-LEVD-fmk, a CASP4-specific inhibitor (Yabuta 2020) [18]. ➢ GGA (20 µM) induced rapid and transient activation of CASP4 from 1 h to 5 h (Yabuta 2020) [18].	➢ PA (300 µM) induced nuclear NF-κB activity in HepG2 cells in 15 min (Joshi-Barve 2007) [65]. ➢ PA (400 µM) induced nuclear translocation of NF-κB in HepaRG cells in 40 min (Sharifnia 2015) [66]. ➢ PA (400 µM) increased mRNA and protein expressions of inflammasome marker NLRP3, CASP1, and IL-1β, as well as GSDMD after 24 h in HepG2 cells (Zeng 2020) [46]. ➢ PA (1000 µM) in normal human liver LO2 cells for 24 h induced activation of CASP1, release of IL-1β, and GSDMD-NT (Yao 2022) [80].	➢ ATRA (10 µM) increased the content of Fe^2+^ in both HepG2 and Hep3B cells (Sun 2022) [51] (evidence for ferroptosis). ➢ ATRA (10 µM) downregulated the expression of GPX4 and FTH1 proteins in both HepG2 and Hep3B cells (Sun 2022) [51] (evidence for ferroptosis). ➢ ATRA (5 µM) upregulated one of the ATRA-responsive genes, OTUD7B, which inhibits NF-κB activity in HepG2 and HuH-7 cells in 6 h (Kanki 2013) [49] (negative evidence for pyroptosis).
Cytoplasmic Ca^2+^	➢ GGA (10 µM) induced a transient increase of cytoplasmic-free Ca^2+^ in HuH-7 cells at 20 min and 6 h (Yabuta 2020) [18].	➢ PA (500–1000 µM) induced lysosomal Ca^2+^ release to increase cytosolic Ca^2+^ in HepG2 cells in 6 h, which was suppressed by cotreatment with MitoTEMPO, a mitochondrial ROS quencher (Oh 2023) [48].	➢ ATRA (1–10 µM) increased [Ca^2+^]i from 24 to 48 h in HepG2 cells (Wei 2014) [67].
LysoPLs	➢ GGA (5 -10 µM) induced in HuH-7 cells, ➢ upregulation of a group of lysophospholipids including lysophosphatidylcholine (lysoPC) with C16:0, C20:4, or C20:3 fatty acids in 24 h (Shidoji 2021) [21]. ➢ the most rapid (2 h) upregulation of lysoPC (C20:4) and lysophosphatidylethanolamine (C20:4) (Shidoji 2021) [21].	➢ lysoPC (40 µM) induced cell death in human primary hepatocytes in 24 h to the same extent as 400–800 µM PA-induced cell death (Han 2008) [71]. ➢ lysoPC (42.5–85 µM) induced cell death both in HuH-7 and human primary hepatocytes in 16 h to the same extent as 800 µM PA-induced cell death (Kakisaka 2012) [81]. ➢ lysoPC (40–80 µM) induced cell death in HepG2 and HuH-7 cells in 18 h (Chen 2022) [82].	✧ Liver lysoPC (16:1 or 18:1) was decreased by oral administration (10 mg/kg, 1 wk) of a synthetic ligand (AM580) for RAR to mice and increased by oral administration (30 mg/kg, 1 wk) of a synthetic ligand for RXR (LG268) to female C57BL6 mice (Weiss 2011) [72].
Autophagy	➢ GGA (10 µM) induced in HuH-7 cells, − massive accumulation of GFP-LC3-labeled autophagosomes (Okamoto 2011) [16]. − time-dependent accumulation of LC3β-II and p62/SQSTM (Okamoto 2011) [16], which was not prevented by cotreatment with 4μ8C (IRE1 Inhibitor III) that inhibits *XBP1* mRNA splicing (Iwao 2015) [17]. − upregulation of ATG4B and BECN1 in 30 min (Okamoto 2011) [16]. − nuclear translocation of cytoplasmic p53 in 3 h (Iwao 2014) [64].	➢ PA (500 µM) impaired autophagic flux indicated by accumulation of LC3β-II and p62/SQSTM through downregulation of DDX58 in HepG2 cells (Frietze 2022) [73]. ➢ PA (500–1000 µM) induced dysfunction of lysosomes in HepG2 cells in 6 h (Oh 2023) [48]. ✧ PA (100 µM) reduced autophagic flow in embryonic mouse hypothalamus cell line N43/5 (Hernandez 2019) [77].	➢ ATRA (40 µM) promoted autophagy in PLC/PRF/5 and HLE cells as demonstrated by, ➢ up-regulation of LC3-II and decrement of p62/SQSTM (Wang 2021) [50]. ➢ up-regulation of ATG7 protein and *ATG7* mRNA (Wang 2021) [50]. ➢ The downregulation of ATG7 gene expression by siRNA enhanced 40 µM ATRA-induced cell death in hepatoma cells (Wang 2021) [50].
CASP3	➢ Cell death induced by 10 µM GGA was delayed by pre-treatment with acDEVD-CHO, a specific inhibitor against CASP3 (Shidoji 1997) [20].	➢ PA (500 µM) increased activation of CASP3 in mouse hepatocyte-derived AML12 cells and mouse primary hepatocytes in 24 h (Huang 2023) [58].	➢ ATRA (40 µM) induced activation of CASP3 activity in PLC/PRF/5 and HLE cells in 24 h (Wang 2021) [50].

➢ Human hepatoma-derived cell lines or rodent hepatocytes. Unless otherwise noted in the GGA section, all observations were obtained using human hepatoma cell lines such as HuH-7, PLC/PRF5, HepG2, and Hep3B. ✧: Cell lines other than those derived from hepatoma or rodent liver.

Due to the information overload in Table 1, it has been boldly simplified and summarized in Table 2 to aid conceptual understanding.

## 4. GGA as an Anti-Oncometabolite

Thus far, we have summarized the mechanism of GGA-induced cell death in HCC cells from the perspective of pyroptosis. Pyroptosis has traditionally been considered a cell-lytic and highly inflammatory form of regulated cell death, particularly studied in immune cells such as macrophages, serving as cell death that activates the innate immune system [83,84,85]. However, studies have increasingly implicated pyroptosis as a major mechanism in the hepatocyte lipotoxicity induced by cholesterol and saturated fatty acids, highlighting the sterile pyroptosis mediated by these endogenous lipids as a mechanism of hepatocyte injury [80,86].

From our studies on HCC prevention, we found that GGA, classified as an acyclic retinoid, induces cell death via pyroptosis in HCC cell lines [18]. Furthermore, we demonstrated that GGA is an endogenous lipid de novo synthesized in the liver from mevalonic acid rather than a vitamin-like acyclic retinoid [11,87]. Therefore, in this review, we have discussed the mechanism of cell death induction by endogenous lipid GGA, comparing it with saturated fatty acid PA and retinoid ATRA. As a result, GGA, like PA, induces cell death in HCC cells via pyroptosis, which is distinctly different from ATRA-induced apoptosis.

While this review has predominantly focused on data obtained from in vitro experiments regarding GGA, there are intriguing observations from in vivo experiments with GGA. In C3H/HeN male mice, the liver GGA content begins to decrease from 7 months of age, decreases further by around 15 months of age, and becomes undetectable by 23 months of age [88]. By this time, almost all mice can be detected with spontaneous liver cancer [89,90]. However, when GGA or 4,5-didehydroGGA is administered orally as a single dose (50 µg/mouse) at 11–12 months of age, the detection rate of liver cancer at 24 months of age is significantly suppressed [88,90]. Since tumors are observed in the liver of C3H/HeN male mice from the first year of life [89], it is conceivable that GGA administered at this time induces pyroptotic cell death in tumor cells, sweeping them away from the liver. In other words, GGA administered at this time may prevent the onset of liver cancer by plucking out its buds [11].

Metabolites that abnormally accumulate in cancer cells and promote tumorigenesis are called oncometabolites, and substances related to the tricarboxylic acid cycle such as 2-hydroxyglutarate, succinate, and fumarate are known examples [91]. These oncometabolites contribute to cancer-specific signatures such as the epigenetic remodeling of tumor cells, abnormal DNA repair, and redox imbalance. On the other hand, although GGA is synthesized as a metabolite derived from mevalonic acid, in certain strains of mice, the liver GGA content decreases with age, and eventually, liver cancer spontaneously occurs. As already mentioned, GGA does not induce pyroptotic cell death or other cell death in normal hepatocytes [20]. Therefore, if the hepatic GGA content does not decrease with age, physiological hepatic GGA induces pyroptotic cell death only in tumor cells that are in the process of carcinogenesis and suppress the development of HCC. Compounds like GGA could be termed anti-oncometabolites (or tumor-suppressive metabolites).

## 5. Conclusions

Initially studied as one of the acyclic retinoids with HCC preventive effects [92], the mechanism of cell death induction in HCC cells by GGA appears to be more like the action of saturated fatty acid PA via UPR^ER^ and mitochondrial dysfunction rather than the action of retinoids mediated by nuclear retinoid receptors [93]. The induction of pyroptosis by PA occurs not only in HCC cells but also to a similar extent in hepatocytes, known as hepatic lipotoxicity [12]. However, the induction of pyroptosis by GGA is selectively observed in HCC cells, and mitochondrial dysfunction or cell death is not observed in the primary hepatocytes after treatment with GGA [20].

How does GGA induce cell death selectively in HCC cells without inducing cell death in hepatocytes? Originally studied in immune cells such as macrophages as an inflammatory form of cell death primarily in response to foreign substances, pyroptosis has recently gained attention as a sterile response in hepatocytes to the abnormal accumulation of endogenous lipids such as cholesterol, saturated fatty acids, and lysophospholipids. GGA, as an endogenous lipid that selectively induces pyroptosis in tumor cells, occupies a slightly different position than saturated fatty acids. If tumor cells are considered foreign substances, further detailed research on the factors determining the selectivity of tumor cells in the liver, such as analyzing changes in the expression levels of the *TLR4* genes accompanying the development of liver cancer [94], is essential for understanding the tumor-selective action of GGA.

When considering the clinical application of these three lipids in the prevention of liver cancer, it will be also important to consider their combined effects.

## Figures and Tables

**Figure 2 cells-13-00809-f002:**
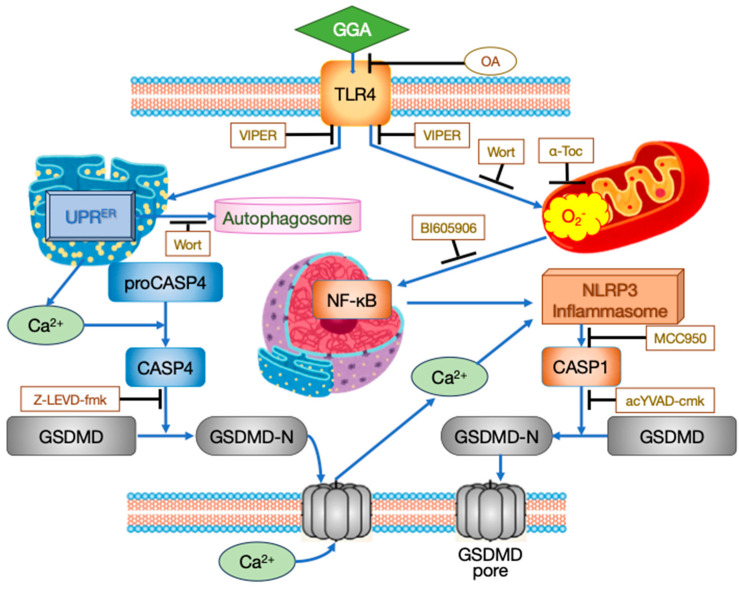
Intracellular processes of GGA-induced cell death in HuH-7 cells. Black T-shaped arrows indicate the site of inhibitor action.

**Figure 3 cells-13-00809-f003:**
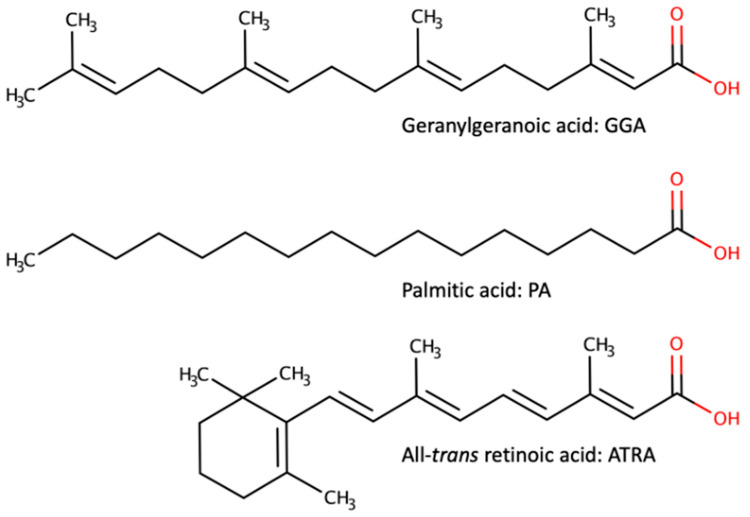
Three fatty acids reported to induce cell death in hepatocellular carcinoma (HCC)-derived cell lines.

**Table 2 cells-13-00809-t002:** Summary of cellular events related to hepatoma cell death after treatment with GGA, PA, or ATRA.

	GGA	PA	ATRA
effective concentration	micromolar	millimolar	micromolar
UPR^ER^	rapid	moderate	rapid
mtROS	very fast	moderate	slow
autophagy	impaired	impaired	enhanced
cytoplasmic Ca^2+^	increased	increased	increased
CASP1	activated	activated	– ^1^
CASP4	activated	– ^1^	– ^1^
CASP3	involved	activated	activated
lysoPLs	increased	increased	decreased
pyroptosis	rapid	slow	– ^1^

^1^ No published papers to date have analyzed.

## Data Availability

The original contributions presented in the study are included in the article/Appendix A, and further inquiries can be directed to the author.

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
