# Peer review of "Induction of Hepatoma Cell Pyroptosis by Endogenous Lipid Geranylgeranoic Acid—A Comparison with Palmitic Acid and Retinoic Acid"

_cells, 2024, doi:10.3390/cells13100809_

Round 1
Reviewer 1 Report
Comments and Suggestions for Authors Yoshihiro Shidoji- PhD
- Principal Investigator at University of Nagasaki
Reviewer 2 Report
Comments and Suggestions for Authors
The review by Shidoji et al., summarises the latest finding on GGA-mediated pyroptotic cell death and comprehensively compares with the cell death induced by palmitic acid and retinoic acid.
The review is very condense and difficult to follow at times. The review is more focused on technical details rather than providing conceptual knowledge. For example, 2.1 where timeline has been described can be summarized in a table or a schematic diagram. A detailed explanation that occur with GGA treatment seem unnecessary and overwhelming.
Similarly, too much details on the concentrations and timing of various treatment. In general reviews should focus on the implications and unique properties rather than technical details.
Throughout the manuscript the authors jump from one mechanism to the other without giving a context. Can the authors summarize or focus on whats similar and different between GGA, ATRA and PA (not focusing on the concentrations or timing of treatment).
GGA inducing pyroptosis should be elaborated and the distinction from PA should be clarified elegantly. As such, it is very confusing whats the author is trying to convey as too many technical details act as a distraction (increasing the content of writing) rather than informative.
If GGA works through TLR4, how GGA selectively kills the cancer cells but not normal hepatocytes. This should be discussed.
Can the authors make a schematic diagram detailing the difference between GGA, PA and ATRA (as in figure 1). Please refrain from using doses and timing details, if such a schematic diagram is provided.
Comments on the Quality of English Language
None
Reviewer 3 Report
Comments and Suggestions for Authors
1. The author compares the mechanism of HCC inhibition by Geranylgeranoic Acid with Palmitic Acid and Retinoic Acid, based on intracellular events occurring in HCC cells after GGA treatment, as reported by different studies and they discuss the inhibition of GGA-induced cell death by various inhibitors. The timeline of intracellular events, thus created, may not be real as these reports use different doses of GGA and varying experimental settings.
2. The author summarizes the inhibition of GGA-induced cell death by various inhibitors, which indicates towards multiple pathways that drive the anti-oncogenic action of GGA.
3. Most of the studies cited by the author are based on in vitro data from cell lines, particularly hepatocytes. However, the title is too generalized to reflect the same.
4. HCC demonstrates an escalated high rate of glycolysis. How does GGA affect glycolysis in HCC cells?
5. Section 3.4, line 424-425, the author states that ‘the effects of the three fatty acids on human HCC cells can be broadly categorized into two types: pyroptosis induced by GGA and PA, and apoptosis 425induced by ATRA’. However, several studies have reported the induction of apoptosis by PA in hepatocytes.
6. The authors need to add if there are any specific pathways or receptors targeted by geranylgeranoic acid that are distinct from those targeted by palmitic acid and retinoic acid
7. Please add if any synergistic effects observed when combining geranylgeranoic acid with palmitic acid or retinoic acid in inhibiting hepatocellular carcinoma growth?
8. Please add the effects of these endogenous lipids on the tumor microenvironment, including immune cell infiltration, cytokine secretion, and angiogenesis, and how these effects influence hepatocellular carcinoma progression?
Round 2
Reviewer 2 Report
Comments and Suggestions for Authors
The authors have addressed my comments. Especially, I liked the timeline representing various cellular events.
Reviewer 3 Report
Comments and Suggestions for Authors
The manuscript has improved.